# Application of Precision Medicine Concepts in Ambulatory Antibiotic Management of Acute Pyelonephritis

**DOI:** 10.3390/pharmacy11060169

**Published:** 2023-10-24

**Authors:** Morgan Pizzuti, Yuwei Vivian Tsai, Hana R. Winders, Paul Brandon Bookstaver, Majdi N. Al-Hasan

**Affiliations:** 1Department of Pharmacy, Prisma Health-Midlands, Columbia, SC 29203, USA; hana.winders@prismahealth.org (H.R.W.); bookstaver@cop.sc.edu (P.B.B.); 2The Johns Hopkins Hospital, Baltimore, MD 21287, USA; ytsai30@jh.edu; 3Department of Clinical Pharmacy and Outcomes Sciences, University of South Carolina College of Pharmacy, Columbia, SC 29208, USA; 4Department of Internal Medicine, Division of Infectious Diseases, Prisma Health-Midlands, Columbia, SC 29203, USA; majdi.alhasan@uscmed.sc.edu; 5Department of Internal Medicine, University of South Carolina School of Medicine, Columbia, SC 29209, USA

**Keywords:** acute pyelonephritis, antibiotic resistance, outpatient antibiotic stewardship, transitions of care, precision medicine

## Abstract

Acute pyelonephritis (APN) is a relatively common community-acquired infection, particularly in women. The early appropriate antibiotic treatment of this potentially life-threatening infection is associated with improved outcomes. The international management guidelines for complicated urinary tract infections and APN recommend using oral antibiotics with <10% resistance among urinary pathogens. However, increasing antibiotic resistance rates among *Escherichia coli* and other Enterobacterales to fluoroquinolones, trimethoprim-sulfamethoxazole (TMP-SMX), and beta-lactams has left patients without reliable oral antibiotic treatment options for APN. This narrative review proposes using precision medicine concepts to improve empirical antibiotic therapy for APN in ambulatory settings. Whereas resistance rates to a particular antibiotic class may exceed 10% at the population-based level, the predicted antibiotic resistance rates based on patient-specific risk factors fall under 10% in many patients with APN on the individual level. The utilization of clinical tools for the prediction of fluoroquinolones, TMP-SMX, and third-generation cephalosporin resistance improves the ambulatory antibiotic management of APN. It may also reduce the need to switch antibiotic therapy later based on the in vitro antibiotic susceptibility testing results of bacterial isolates in urinary cultures. This approach may mitigate the burden of increasing antibiotic resistance in the community by ensuring that the initial antibiotic prescribed has the highest likelihood of treating APN appropriately.

## 1. Introduction

Urinary tract infections (UTIs) represent one of the most common outpatient infections treated with antibiotics, primarily affecting young women. *Escherichia coli* is the predominant bacteria in both uncomplicated cystitis and acute pyelonephritis (APN), followed by other Enterobacterales such as *Klebsiella* spp. and *Proteus mirabilis*. In a population-based study over a 5-year period, the annual rate of outpatient pyelonephritis among the female population in the United States was 12 to 13 cases per 10,000 people [1]. The management of APN in outpatient settings necessitates the use of highly bioavailable oral antibiotics obtaining high urine concentrations, such as fluoroquinolones or trimethoprim-sulfamethoxazole (TMP-SMX). While the incidence of APN remains relatively stable, the rates of antibiotic resistance to both fluoroquinolones and TMP-SMX are increasingly evident [1,2]. Additionally, the emergence of extended-spectrum beta-lactamase (ESBL)-producing Enterobacterales in the community poses a significant concern due to the common cross-resistance to fluoroquinolones and TMP-SMX among these isolates [2].

With the increase in antibiotic resistance rates among community-onset urinary tract isolates, it is imperative to consider additional treatment strategies. The goal of this proposed strategy is to improve the selection of antibiotic therapy in individuals with APN and reduce the likelihood of antibiotic resistance transmission at the population level. This narrative review outlines key perspectives and considerations of applied precision medicine as a tool to manage APN when the selection of the antibiotic regimen is stratified to the host, microbiome, and pathogen characteristics.

## 2. The 2010 Infectious Diseases Society of American (IDSA) and European Society of Clinical Microbiology and Infectious Diseases (ESCMID) Guidelines on APN

The 2010 IDSA and ESCMID guidelines recommend the use of oral ciprofloxacin, levofloxacin, or TMP-SMX for the management of APN [3]. These guidelines do not provide separate frameworks of recommendations for inpatient and outpatient settings. The empiric use of oral fluoroquinolones is recommended when the prevalence of fluoroquinolone resistance is <10%, whereas the use of TMP-SMX is only encouraged when in vitro susceptibility is known. When the prevalence of community fluoroquinolone resistance exceeds 10% and a fluoroquinolone is used empirically, an initial dose of a long-acting parenteral agent, such as ceftriaxone or an aminoglycoside, is recommended. An initial dose of a long-acting parenteral agent is also recommended if TMP-SMX is being used empirically. The use of oral beta-lactams may also be considered for a prolonged treatment duration, but they are noted to be less effective than fluoroquinolones and TMP-SMX due to reduced penetration into the site of infection in the kidneys. Therefore, an initial parenteral dose of ceftriaxone or aminoglycoside should be considered in this scenario. The duration of therapy ranges between 5 and 14 days depending on the antibiotic due to differences in bioavailability and tissue penetration. Due to the low prevalence in the community at the time when the guidelines were written, no specific antibiotic regimens were recommended for empiric therapy in the setting of APN in individuals with a high risk of infections due to ESBL-producing Enterobacterales.

## 3. Prevalence of Antibiotic Resistance

Studies from the early 2000s demonstrated that the rate of *E. coli* resistance to fluoroquinolones in the United States was <7%, although certain regions (e.g., Southwest) had an elevated resistance rate of up to 20% [4]. In a more recent cross-sectional study of 10 emergency department (ED) sites surveyed, the prevalence of fluoroquinolone resistance increased substantially. with all sites having a rate of >10% [2]. For TMP-SMX resistance, the prevalence in the United States is at least 20% and up to 40% based on more recent surveillance data, although healthcare-associated UTIs were also considered in this study [4,5]. Similar trends in antibiotic resistance in Europe are also noted. Studies evaluating antimicrobial resistance among urinary isolates from Europe demonstrated resistance rates of 30% for both fluoroquinolones and TMP-SMX [6,7]. While ESBL-producing Enterobacterales have traditionally been of a low concern for community-acquired UTIs, data have emerged recently showing an increasing prevalence (up to 12%) owing to the globally disseminated, multidrug-resistant clone, ST131 *E. coli*, that accounts for 85% of these infections through the production of CTX-M-15 beta-lactamases [2].

## 4. Application of Precision Medicine Concept in APN

### 4.1. Summary of Concept

Precision medicine applied in infectious diseases is a relatively new concept that proposes the customization of therapeutics and medical decisions based on an individual’s clinical and environmental characteristics rather than a one-size-fits-all approach [8]. When selecting an empiric antibiotic regimen, precision medicine leverages the host’s risk factors along with the local antibiotic susceptibility patterns to provide the best predicted patient-specific antibiogram. The use of local antibiograms and resistance risk factor prediction tools to deliver the most effective treatment regimen is a sentiment echoed throughout nearly all infectious syndromes. This is especially important when prescribing antibiotics in the outpatient setting, where timely culture and susceptibility reports are not always feasible. Therefore, the development of local antibiograms and the verification of risk factors for antibiotic resistance specific to the patient level are imperative to optimize the appropriate use of antibiotics.

### 4.2. Resistance Rates in the United States

In a study by Dunne and colleagues that included 15 institutions in the United States, based on urinary isolates of Enterobacterales from over 5000 patients, investigators found resistance rates of >20% to fluoroquinolones, TMP-SMX, and beta-lactam antibiotics [9]. This study compared outpatient prescriptions, the resistance of pathogens to those prescriptions, and the frequency of a second prescription. Of note, 22% of patients were prescribed an antibiotic that the urinary pathogen was resistant to. These patients were twice as likely to receive an additional antibiotic prescription or be hospitalized within 28 days of the initial prescription. The authors found that the risk of treatment failure, outside of having a resistant pathogen to the antibiotic prescribed, included an age greater than 60 years old, diabetes mellitus, and male sex.

While the selection of an empiric antibiotic regimen is often guided by the appropriate minimum acceptable susceptibility (MAS) based on a local antibiogram, other patient-specific markers should also be factored into the threshold of MAS. Haggard and colleagues demonstrated a proportional increase in the median MAS target among a cohort of clinically trained pharmacists with an increasing severity of Gram-negative infections and poor clinical prognoses. Specifically, the median MAS for critically ill patients with bloodstream infections and a poor or guarded prognosis was 90%, whereas the MAS for patients with APN and a good prognosis was 85% [10]. Another study focused on empiric antibiotic selection for APN with a proposed algorithm that incorporated multiple patient risk factors (e.g., risks for antibiotic resistance and severity of illness) in determining the MAS. In particular, the MAS might be 85% for a healthy patient with uncomplicated APN, as opposed to 90% in a patient with complicating factors [8]. This supports the establishment of a patient-specific MAS, which allows for flexibility in determining the most appropriate empiric antibiotic regimen for each individual patient.

### 4.3. Institutional Antibiograms

Establishing an institutional antibiogram may not always be feasible, especially at smaller hospitals where there may be a lack of personnel and resources. Similarly, the outpatient setting also lacks resources to complete an antibiogram. Non-variable challenges include few pathogen isolates and therefore do not meet the CLSI standard of 30 isolates per species. Microbiologic testing may not be widely available in rural areas or may be limited to send out testing, creating a barrier to the primary source data (i.e., Vitek, etc.). Once an antibiogram is completed, if not appropriately analyzed and distributed, it may send the wrong message to prescribers and influence antibiotic ordering incorrectly. There may be a lack of knowledge on interpreting and utilizing the antibiogram effectively. Education is key for prescribers to enact change in prescribing patterns based on an antibiogram.

If a hospital chooses to complete an institutional antibiogram, ensuring correctness is of utmost importance. Upon a review of 37 hospitals in the Duke Antimicrobial Stewardship Outreach Network (DASON) group, the authors found that 25% of hospitals reported antibiograms with only >30 isolates per species. Additionally, CLSI required standards were observed correctly in only 9% of antibiograms [11]. If a hospital is concerned with the number of isolates needed to meet CLSI standards, an alternative approach for smaller community hospitals is to produce a regional antibiogram for more accurate prescriber practices [12]. Another alternative approach is to use a longer period for susceptibility analysis.

### 4.4. Problems with Antibiograms

While an antibiogram is an important antimicrobial stewardship tool that can aid in the selection of empiric antibiotic regimens, there are limitations with the applicability of traditional antibiograms. Most hospitals typically distribute a general antibiogram that includes isolates from all sources and all units, which may be difficult to interpret as different units may have different resistance rates owing to the complexity of patients served (i.e., intensive care units versus medical ward units). Secondly, antibiograms do not distinguish between community-acquired versus hospital-acquired infections, whereby the latter are generally associated with higher rates of resistance. Thirdly, outpatient physician offices do not have clinic-specific antibiograms available for patients that have routine visits. Fourthly, a traditional antibiogram may include colonizing organisms, which could influence the susceptibility rates. Fifthly, the distribution of quantitative data such as the minimum inhibitory concentration (MIC) is not represented by the antibiogram, which is pertinent for the optimization of antibiotic dosing, especially in individuals with altered pharmacokinetic parameters (e.g., extreme body habitus and augmented renal clearance). Lastly, static antibiograms may be of limited use in empiric antibiotic selection in patients with recurrent or recent infections.

### 4.5. Institutional- vs. Unit-Based Antibiograms

Pathogen susceptibility may be incorrectly interpreted when isolates from all units within an institution are combined as a cumulative antibiogram. Pathogens recovered from intensive care units (ICUs) typically have significantly higher rates of resistance compared to those from non-ICU locations due to the inherent risk for the repeated, prolonged exposure to broad-spectrum antibiotics and the severity of illness [13,14]. Therefore, the selection of empiric antibiotic regimens based on the hospital’s cumulative antibiogram could potentially underestimate the rate of resistance for patients in ICU locations, resulting in the use of suboptimal antibiotic agents.

### 4.6. All-Source versus Urine-Specific Antibiograms

Creating a urine-specific antibiogram may be helpful for determining empiric therapy, especially in outpatient and ED settings. In a study by Rabs and colleagues, 2284 pathogens were isolated from a urinary source. Overall, the antibiotic susceptibility increased; however, the prevalence of ESBL *E. coli* isolates was greater in the urinary antibiogram versus the standard (13% versus 9%, *p* < 0.001) [15]. In a veteran population of 2494 urine isolates that were analyzed, the antibiotic resistance for fluoroquinolones and TMP-SMX significantly differed when comparing *E. coli* isolates alone versus all urinary isolates (28% vs. 39%, *p* < 0.001) [16]. These findings demonstrate the potential utility of a urine-specific antibiogram.

### 4.7. Population Level vs. Patient-Specific Antibiograms

Contrary to the traditional antibiogram, a patient-specific antibiogram accounts for numerous variables that are known to impact a patient’s risk of having a resistant organism. Overly and colleagues demonstrated that a patient-specific antibiogram based on the inclusion of key elements, including prior antibiotic exposure, prior microbiology, a history of resistant pathogens, diagnoses, age, and location in the hospital, provided a better prediction of antibiotic susceptibility compared to the traditional antibiogram [17]. More importantly, the antimicrobial resistance rates to all first-line oral options for APN have exceeded 20% in institutional antibiograms in most parts of the world. This makes population-level antibiograms obsolete in the selection of empiric oral antimicrobial therapy for APN.

## 5. Prediction of Antibiotic Resistance in Enterobacterales

### 5.1. Prediction of Fluoroquinolone Resistance

Two single-center studies conducted in the United States examined independent risk factors for UTIs due to fluoroquinolone-resistant bacteria. Shah and colleagues identified multiple variables associated with fluoroquinolone resistance including the male sex, residence at a skilled nursing facility, an outpatient procedure within 1 month of the index infection, and prior fluoroquinolone use within 3 months and within 3 to 12 months of the index infection [18]. In addition, Rattanaumpawan and colleagues identified other variables that could contribute to fluoroquinolone resistance including a recent exposure to metronidazole or TMP-SMX, medicine service, hydronephrosis, and renal insufficiency [19].

### 5.2. Prediction of TMP-SMX Resistance

The independent risk factors predicting TMP-SMX resistance in UTIs have been reported by several studies [20,21]. In a retrospective single health system review that included primarily females (72%) with *E. coli* community-acquired complicated UTIs (61%), the authors found that a prior UTI or urinary colonization with TMP-SMX resistant isolates and prior use of TMP-SMX within the past 12 months of the index infection were associated with a risk of TMP-SMX resistance [20]. Another study examined seven years of urine culture data collected from 574 female patients who presented to primary care clinics with uncomplicated UTIs due to *E. coli* in Michigan. The findings suggested that the recent use of TMP-SMX was associated with a higher risk of TMP-SMX resistance than the recent use of any other antibiotic (OR 16.74 [95% CI 2.90–96.95] vs. OR 2.37 [1.14–4.95], respectively) [22]. Another retrospective case–control study of patients with community-acquired UTIs due to *E. coli*, recruited from a Veterans Affair ambulatory clinic, also found that prior exposure to antibiotics including TMP-SMX, fluoroquinolones, and tetracyclines within the past 6 months increased the risk of TMP-SMX resistance. Furthermore, the study also demonstrated that as the number of previous antibiotic courses increased, the frequency of TMP-SMX resistance also increased. This ranged from a TMP-SMX resistance of 8% to 39% for patients with no prior antibiotics vs. patients with three or more prior antibiotics, respectively [21].

### 5.3. Prediction of Ceftriaxone Resistance (ESBL-Production)

The increasing rates of ESBL-producing Enterobacterales detected in the community is concerning to all clinicians. In a large, multicenter, retrospective case–control study of adult patients admitted with community-acquired UTIs due to ESBL-producing Enterobacterales, those with a history of repeated UTIs, the presence of a urinary catheter at the time of admission, and prior exposure to outpatient antibiotics within the past 3 months had an increased risk for UTI due to ESBL-producing Enterobacterales after controlling for the acute severity of illness and comorbid conditions [23]. Another study examined the risk factors for ESBL-production among patients with a bloodstream infection (BSI) due to Enterobacterales. Notably, the majority of patients had a urinary source of infection and 79% had community-onset BSI. The risk factors for BSI due to ESBL-producing Enterobacterales included prior beta-lactam and/or fluoroquinolone use within the past 90 days, an outpatient gastrointestinal and/or genitourinary procedure within the past month, and a prior infection and/or colonization with an ESBL-producing organism within the past 12 months [24].

In a study by Goodman and colleagues, a clinical risk score with 14 variables was compared to a 5-variable decision tree model to predict patients’ risk of ESBL bacteremia. While this study focused on BSI, the authors found that the decision tree model, which included variables such as a history of infection due to ESBL-producing bacteria, local ESBL rates, indwelling hardware, prior antibiotic courses, and age, was more user-friendly but had lower discrimination when compared to the clinical risk score. This is important to consider when developing a risk score tool for the empiric selection of antibiotics based on predicted resistance [25].

### 5.4. Potential Benefits of Patient-Specific Therapy

A patient-specific risk factor evaluation would lead to a higher likelihood of selecting the appropriate antibiotic agent upon the initial patient presentation in the ambulatory setting. This would likely lead to faster clinical improvement in patients with APN. It may also result in a reduced risk of clinical deterioration, ultimately leading to hospitalization. This approach would also reduce the likelihood of future antibiotic switches that may be required if the bacterial isolate in the urine was resistant to the prescribed antibiotic. This will reduce the risk of exposure to multiple classes of antibiotics for the treatment of the same infection, which may subsequently reduce the risk of colonization with multi-drug resistant pathogens and *Clostridioides difficile* infection. Frequent switches in ambulatory antibiotic prescriptions may also lead to a reduction in patients’ compliance with therapy due to cost and other logistical barriers to the second prescription. In addition, multiple antibiotic prescriptions may lead to a higher likelihood of developing an adverse drug reaction.

As demonstrated in the study by Dunne and colleagues, patients with resistant pathogens often need an alternative prescription, which places strain on the healthcare system [9]. Another US study conducted by Jorgensen and colleagues analyzed the rate of emergency department return visits (ERVs) in relation to UTIs [26]. In this study, 15% of patients had an ERV, and 47% of those were related to a UTI. Of note, among the patients that had an ERV, 47% had a resistance to the discharge antibiotic. Once again, this study reiterates the burden of incorrectly stratifying patients in a one-size-fits-all category for antibiotic prescribing. This can lead to return primary care visits, return ED visits, or even hospitalization. It is of utmost importance to ensure that patients are receiving the correct antibiotic based on patient-specific risk factors from the beginning of their treatment course to prevent future complications.

## 6. Clinical Application of Guideline-Based vs. Precision-Medicine-Based Empirical Therapy for APN (Figure 1 and Figure 2; Table 1, Table 2 and Table 3)

Figure 1 shows the Guidelines-Based Therapy of Acute Pyelonephritis.

**Figure 1 pharmacy-11-00169-f001:**
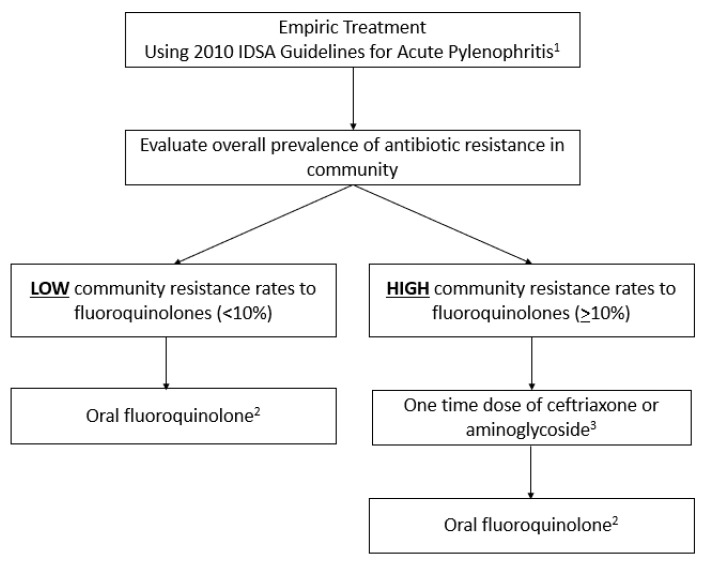
Guideline-based therapy of acute pyelonephritis [3]. ^1^ [3]. ^2^ Oral ciprofloxacin 500 mg twice daily for 7 days or levofloxacin 750 mg daily for 5 days in patients not requiring hospitalization. ^3^ IV Ceftriaxone 1G or consolidated 24 h dose of an aminoglycoside. For SMX-TMP: not recommended unless susceptibilities are known. If TMP-SMX is used empirically, an initial dose of IV ceftriaxone 1G or consolidated 24 h dose of aminoglycoside is recommended. Oral beta-lactam agents are less effective than other available agents. If an oral beta-lactam is used empirically, an initial dose of IV ceftriaxone 1G or consolidated 24 h dose of aminoglycoside is recommended. For beta-lactams: duration of 10–14 days.

**Figure 2 pharmacy-11-00169-f002:**
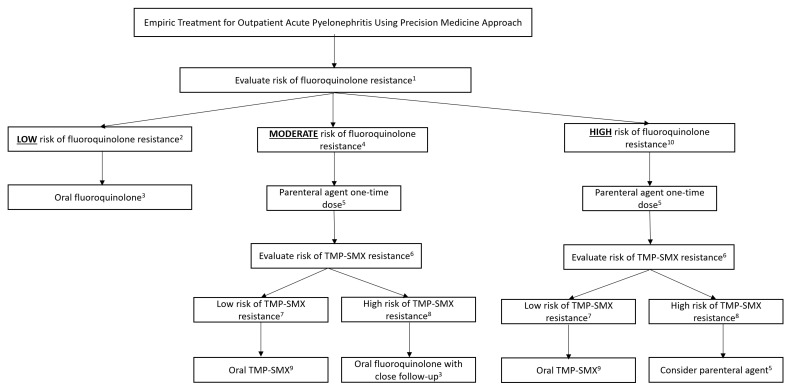
Precision-medicine-based therapy of acute pyelonephritis. ^1^ See fluoroquinolone risk score in Table 1. ^2^ Low risk of fluoroquinolone resistance defined as fluoroquinolone resistance score < 2. ^3^ Levofloxacin 750 mg PO Q24 h or ciprofloxacin 500 mg PO Q12 h for 5–7 days. ^4^ Moderate risk of fluoroquinolone resistance defined as risk score of 2. ^5^ Parenteral agents to include: IV/IM ceftriaxone 1G or consolidated 24 h dose of tobramycin or amikacin. ^6^ See TMP-SMX risk score in Table 2. ^7^ Low risk of TMP-SMX resistance defined as risk score < 1. ^8^ High risk of TMP-SMX resistance defined as risk score ≥ 1. ^9^ 1 Double-strength TMP-SMX PO Q12 h × 14 days. ^10^ High risk of fluoroquinolone resistance defined as fluoroquinolone resistance score > 2. If there is a high risk of fluoroquinolone resistance and a high risk of TMP-SMX resistance, consider evaluating for risk of ESBL (Table 3) and *Pseudomonas aeruginosa*. If there is concern for *Pseudomonas aeruginosa*, consider utilizing anti-pseudomonal agent such as meropenem, piperacillin/tazobactam, cefepime, tobramycin, or amikacin. If ESBL prediction score is <3, and there is no concern for *P. aeruginosa*, utilize IV/IM ceftriaxone. If ESBL prediction score is ≥3, utilize IV/IM ertapenem or an aminoglycoside. Tobramycin or amikacin are the only recommended aminoglycosides due to the recent change in recommendations by CLSI to not utilize gentamicin as a treatment option for *P. aeruginosa* [27].

**Table 1 pharmacy-11-00169-t001:** Fluoroquinolone resistance score.

Patient-Specific Characteristic	Points
Male sex	1
Diabetes mellitus	1
Residence in skilled nursing facility	2
Outpatient procedure in past 30 days	3
Fluoroquinolone use in past 3 months	5
Fluroquinolone use in past 3–12 months	3
**Fluroquinolone Resistance Score Interpretation** Score < 2 implies < 10% predicted probability of fluoroquinolone resistance Score of 2 implies 15% predicted probability of fluoroquinolone resistance Score ≥ 2 implies > 20% predicted probability of fluoroquinolone resistance	
*May utilize local validated risk scores if applicable with corresponding risk cutoffs*	
Adapted from [18]	

**Table 2 pharmacy-11-00169-t002:** Trimethoprim–sulfamethoxazole risk score.

Patient-Specific Characteristic	Points
TMP-SMX use in past 12 months	1
Prior urine cultures with TMP-SMX-resistant pathogen in past 12 months	2
**TMP-SMX Risk Score Interpretation** Score < 1 implies < 13% predicted probability of TMP-SMX resistance Score ≥ 1 implies > 30% predicted probability of TMP-SMX resistance	
*May utilize local validated risk scores if applicable with corresponding risk cutoffs*	
Adapted from [20]	

**Table 3 pharmacy-11-00169-t003:** Extended-spectrum beta-lactamase (ESBL) prediction score.

Patient-Specific Characteristic	Points
Outpatient GI/GU procedure in past 30 days	1
One beta-lactam or fluoroquinolone courses within past 90 days	1
Two or more beta-lactam or fluoroquinolone courses within past 90 days	3
Documented colonization or infections with ESBLs within past 12 months	4
**ESBL Risk Score Interpretation** Score < 3 implies < 10% predicted probability of ceftriaxone resistance for Enterobacterales Score ≥ 3 implies > 20% predicted probability of ceftriaxone resistance for Enterobacterales	
*May utilize local validated risk scores if applicable with corresponding risk cutoffs*	
Adapted from [24]	

## 7. Case Example

A 47-year-old woman is seen at an outpatient primary care clinic with complaints of dysuria, increased urinary frequency, and flank pain. The provider orders a urinalysis (UA) with culture, and the UA results are below (Table 4). The patient has a past medical history of diabetes mellitus type 2, hyperlipidemia, and hypertension. The patient has not used any antibiotics in the past year. Of note, the provider references the local antibiogram for a nearby hospital and finds that *Escherichia coli* has a susceptibility to ciprofloxacin and levofloxacin at 75%.

Allergies: penicillin (hives)

Medication list:Metformin 1000 mg twice daily;Lisinopril 20 mg daily;Atorvastatin 40 mg daily.

### 7.1. Empiric Therapy Using 2010 IDSA Guidelines for APN

Utilizing the 2010 IDSA Guidelines for APN [3], after evaluating the overall prevalence of fluoroquinolone resistance, which is >10%, the provider is faced with utilizing an intravenous one-time dose of either ceftriaxone or aminoglycoside. Due to the overwhelmingly high community resistance, even in the United States, this is often the decision providers are faced with. If the clinic is located in a rural setting with limited access to a timely culture turnaround, this one-time dose may then need to be repeated before the culture’s return. This poses challenges in feasibility for the patient to adhere to this regimen due to logistics and places a burden on the healthcare system. In addition, IV ceftriaxone and/or aminoglycosides can also be accompanied by adverse effects, and the availability of these antibiotics may be limited in an outpatient setting. If the healthcare provider prescribes oral fluoroquinolones following one dose of IV/IM ceftriaxone, there is a 25% chance that the urinary isolate may be resistant to fluoroquinolones. A second antibiotic prescription may be necessary in one-fourth of patients with APN.

### 7.2. Empiric Therapy Using Precision Medicine Approach

Utilizing the precision medicine approach, the risk scores are relied on to direct the provider to the most appropriate antibiotic choice. First, the probability of fluoroquinolone resistance must be evaluated in this particular patient. Based on a fluoroquinolone resistance score of <2 (one point for diabetes mellitus), the predicted risk of fluoroquinolone resistance in this patient is <10%. This would place the patient in the category of “low risk of fluoroquinolone resistance”. Accordingly, there will be no need for the administration of IV or IM ceftriaxone or an aminoglycoside. This will spare the patient from additional costs, potential antibiotic adverse effects, and infusion-related complications. The patient may be prescribed oral ciprofloxacin or levofloxacin. The probability that the patient will need a second antibiotic prescription due to the growth of a fluoroquinolone-resistant urinary bacteria is <10%.

## 8. Applications in Pharmacy Practice

Pharmacists play integral roles in antimicrobial stewardship in the community and large academic hospitals. The application of precision medicine would largely benefit the antimicrobial stewardship and ambulatory care pharmacists in optimizing empiric treatment for the management of APN. Integrating guidelines with the technology that is already implemented at institutions is necessary to allow for utilization by prescribers. A few ways to implement these risk factor tools include smart phone applications, as well as embedding them into the drug ordering pathway itself within the electronic health record (EHR) system. Disseminating these risk factor tools and education to prescribers will increase the utilization of local guidelines and risk factor prediction models. Pharmacists are well equipped to deliver this education to prescribers of all stages including attendings and trainees.

In a study conducted by Wang and colleagues at a 1400 bed tertiary hospital in China, interventions made by antimicrobial stewardship pharmacists had tremendous impacts on antibiotic consumption and antibiotic resistance. The authors found that the number of antibiotic prescriptions in the outpatient and inpatient settings decreased, and the amount of antibiotic prophylaxis also decreased. Pharmacists also play key roles in the appropriate dosing and timing of antibiotic administration in the hospital setting. Along with a decrease in antibiotic consumption, the study demonstrated that the fluoroquinolone resistance rates of *E. coli* isolates decreased by 1.6% for levofloxacin and 1.4% for ciprofloxacin each year (*p* < 0.01 and *p* < 0.001) [28].

While the value of antimicrobial stewardship pharmacists in the inpatient, hospitalized setting is becoming well known and attributable to patient outcomes, the outpatient setting may be an untapped area for potential opportunities. In a cross-sectional, multicenter survey study conducted by Eudy and colleagues, the authors demonstrated that antimicrobial stewardship pharmacists were present in 7% (9 of 129) of ambulatory settings. This was compared to an 88% (114 of 129) presence in hospital settings [29]. This demonstrates a need for pharmacists to drive antibiotic use, specifically in the outpatient setting.

## 9. Limitations

There are several limitations to this narrative review. First, articles examining the risk factors for fluoroquinolone-, TMP/SMX-, or ceftriaxone-resistant uropathogens are based on single-center retrospective studies. Therefore, the results are limited by its lack of external validation. Secondly, this paper does not address the outpatient management of APN in pregnant patients. The use of fluoroquinolones and trimethoprim–sulfamethoxazole is generally discouraged in pregnancy [30]. The guideline endorsed by the American College of Obstetricians and Gynecologists should be considered for treatment approach in this particular patient population [31]. Lastly, the proposed recommendations presented are for the clinical diagnosis of APN only, and are not appropriate for the management of UTIs with complicating factors such as urinary tract obstruction.

## 10. Summary

In conclusion, the application of patient-specific precision medicine algorithms, such as the one proposed in this article, will optimize the ambulatory antibiotic management of APN. Providing the most effective, optimal therapy while minimizing consequences, such as side effects and further resistance, is crucial to patient care and antimicrobial stewardship initiatives. This algorithm provides a framework for healthcare systems to develop institution-specific protocols for the ambulatory antibiotic management of APN based on local antibiotic resistance data. Targeting patient-specific risk factors allows for a patient-centered approach to medicine, which is encouraged throughout the 2010 IDSA and ESCMID guidelines on APN. The application of precision medicine concepts improves the appropriateness of the first ambulatory antibiotic prescribed for APN and reduces the probability of clinical deterioration due to delayed appropriate therapy. It also reduces the risks of providing multiple antibiotic prescriptions for the treatment of the same infection and the associated adverse effects from these antibiotics.

## Figures and Tables

**Table 4 pharmacy-11-00169-t004:** Urinalysis for case example.

Characteristic	Result	Reference Range
Color	Yellow	Clear Yellow
Clarity	Cloudy	Clear
LE	Moderate	Negative
Nitrite	Moderate	Negative
Protein	Negative	Negative
WBC	126 HPF	0–5 HPF
RBC	0 HPF	0–2 HPF
Bacteria	Moderate	Negative
LE = leukocyte esterase HPF = cells per high power field		

## Data Availability

Not applicable.

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
