# Peer review of "Application of Precision Medicine Concepts in Ambulatory Antibiotic Management of Acute Pyelonephritis"

_pharmacy, 2023, doi:10.3390/pharmacy11060169_

Round 1

Reviewer 1 Report

Excellent article. The authors did a great job in describing the issues that clinicians face when prescribing  antibiotics in order to treat patients suffering from Acute Pyelonephritis. As antibiotic resistance rates have become an issue worldwide, such efforts that aim to tackle this problem are always welcome. Easy to implement algorithms such as the one suggested by the authors are always welcome. It would be interesting to see the results on the daily clinical practice in the future. 
The use of English language is perfect and the flow is good, thus making the article easy and interesting to read. 

Author Response

Thank you for the comments!

Reviewer 2 Report

Dear Authors,

I read your manuscript entitled "Application of Precision Medicine Concepts in Ambulatory Antibiotic Management of Acute Pyelonephritis" with great interest. Overall, the paper addresses an intriguing and highly relevant issue in modern medicine.

I would like to commend the clear and convenient flowcharts as well as the well-constructed tables proposed.

I would like to suggest adding a brief paragraph about resistance rates in Europe, while also citing the ESCMID for reference.

Additionally, I would like to encourage the authors to discuss the limitations of this narrative review. For instance, it would be valuable to address aspects such as the lack of external validation for the clinical prediction models of antibiotic resistance or the retrospective design of the studies cited.

Best regards

Dear Editor,

Thank you for providing me with the opportunity to revise this study. This article delves into the crucial aspects of applying precision medicine as a means to manage Acute Pyelonephritis (APN) in a more personalized manner, with the aim of minimizing side effects, mitigating antibiotic resistance, and reducing costs.

I propose the publication of the present article following only minor revisions.

Sincerely

Author Response

Thank you for the comments! We have added prevalence data on antibiotic resistance in Europe (line 91-95) as well as adding a limitation section (line 410-422) addressing the limitations of this review article. 

Reviewer 3 Report

A very nice narrative revies regarding the management of APN in outclinic patients. The writing is concise, with a good flow that direct the reader towards a very strong message.

Major issue

1. The material and method section is missing. The authors should introduce a small paragraph describing the databases searched and timeframe for the articles selected for this manuscript. Although this paper is a non systematic review, this section is required in order to increase the transparency of the work of the authors and increase the strength of this review.

 2. The article brings into debate the management of primary acute uncomplicated pyelonephritis caused by bacterial ascent through the urethra and urinary bladder. In order to have this diagnosis the doctor must exclude the presence of urinary tract obstruction (ultrasound/sectional imaging) and hematological dissemination. This diagnosis of exclusion must be discussed by the authors in the case example presented since flank pain may hide a renal colic due to a ureteric stone for example and the management pathway is different. In this case is more likely an ignored cystitis that ascended, but the differential diagnosis is still needed.

3. The management of APN in pregnant women is missing. Since this paper addresses the individualized approach of APN, I think that a small paragraph regarding this subset of patients is needed, since the majority of pregnant women cannot use chinolones or TMX due to possible teratogenic effects.

Minor issues

Small grammar mistakes and typos (example (“urin” page 10, table 2).

some minor grammar mistakes and misspelling (i.e. urin page 10, table 2). 

Author Response

Thank you for the feedback. We do not think a method section is warranted and appropriate for this paper as it is a narrative review not systematic literature review. We have addressed this concern by stating in the abstract and the introduction sections that this is a narrative review. Additionally, we have addressed your concerns about management in pregnancy as well as ruling out complicated UTIs (e.g. obstruction) in the limitation section (line 410-422).

Round 2

Reviewer 3 Report

The authors have addressed all the issues reported in the previous review. No further issues from my part